# Integrative System Biology Analyses Identify Seven MicroRNAs to Predict Heart Failure

**DOI:** 10.3390/ncrna5010022

**Published:** 2019-03-07

**Authors:** Henri Charrier, Marie Cuvelliez, Emilie Dubois-Deruy, Paul Mulder, Vincent Richard, Christophe Bauters, Florence Pinet

**Affiliations:** 1Inserm, Institut Pasteur de Lille, Université de Lille, FHU-REMOD-VHF, U1167-RID-AGE, F-59000 Lille, France; henri.charrier@pasteur-lille.fr (H.C.); marie.cuvelliez@pasteur-lille.fr (M.C.); emilie.deruy@pasteur-lille.fr (E.D.-D.); 2Normandie University, UNIROUEN, Inserm U1096, FHU-REMOD-VHF, 7F-6000 Rouen, France; paul.mulder@univ-rouen.fr (P.M.); vincent.richard@univ-rouen.fr (V.R.); 3Inserm, Institut Pasteur de Lille, Faculté de Médecine de Lille, Université de Lille, CHU Lille, FHU REMOD-VHF, U1167-RID-AGE, F-59000 Lille, France; christophe.bauters@chru-lille.fr

**Keywords:** biomarkers, miRNAs, heart failure, system biology

## Abstract

Heart failure (HF) has several etiologies including myocardial infarction (MI) and left ventricular remodeling (LVR), but its progression remains difficult to predict in clinical practice. Systems biology analyses of LVR after MI provide molecular insights into this event such as modulation of microRNA (miRNA) that could be used as a signature of HF progression. To define a miRNA signature of LVR after MI, we use 2 systems biology approaches, integrating either proteomic data generated from LV of post-MI rat induced by left coronary artery ligation or multi-omics data (proteins and non-coding RNAs) generated from plasma of post-MI patients from the REVE-2 study. The first approach predicted that 13 miRNAs and 3 of these miRNAs would be validated to be associated with LVR in vivo: miR-21-5p, miR-23a-3p and miR-222-3p. The second approach predicted that 24 miRNAs among 1310 molecules and 6 of these miRNAs would be selected to be associated with LVR in silico: miR-17-5p, miR-21-5p, miR-26b-5p, miR-222-3p, miR-335-5p and miR-375. We identified a signature of 7 microRNAs associated with LVR after MI that support the interest of integrative systems biology analyses to define a miRNA signature of HF progression.

## 1. Introduction

Heart failure (HF) is a major cause of mortality in occidental countries that is difficult to predict in clinical practice [1]. HF can be the consequence of left ventricle remodeling (LVR) induced by a myocardial infarction (MI) [2,3]. LVR is characterized by cardiac hypertrophy and reduction of LV wall. Although LVR is an adaptive response early after MI, it is becoming deleterious in the long term [4]. Deciphering molecular events underlying LVR may offer new opportunities in the identification of early predictive biomarkers of LVR and HF. Omics approaches including transcriptomics, proteomics, and metabolomics have been extensively used to explore these mechanisms but the amount of complex generated data prevents their comprehensive analysis.

Recently, systems biology opened new opportunities to better understand molecular networks and identify new targets involved in HF [5,6]. Among those potential targets, microRNAs (miRNAs) are non-coding RNAs of 19 to 23 nucleotides that regulate gene expression by targeting messenger RNAs [7]. MiRNAs are involved in many processes like cardiomyocyte hypertrophy, fibroblast to myofibroblast transformation and cell to cell communication [8,9,10]. The modulation of expression of a small set of miRNAs associated with LVR may define a miRNA signature to detect this process. The present study aims to define a signature of miRNAs associated with LVR after MI to predict HF. We analyzed two systems biology approaches that we previously developed from a post MI rat model [10,11,12,13] and post-MI patients from the REVE-2 study [14].

## 2. Results

### 2.1. Analysis of the Protein-MiRNA Network Derived from Post-MI Rats Identified Circulating MiR-21-5p, MiR-23a-3p and MiR-222-3p to Be Associated with LVR after MI

The proteomic screening in LV of post-MI male rats [11], previously published [12,13], allowed the identification of 45 proteins modulated by LVR. Using the Qiagen’s Ingenuity Pathway knowledge base, we built a protein-miRNA interaction network highlighting 13 candidate miRNAs which were prioritized to identify candidate miRNAs to detect LVR after MI (Figure 1).

The 13 candidate miRNAs are predicted to interact with 8 out of the 45 proteins modulated by LVR [10], testifying that they are involved in LVR after MI. Moreover, 9 out of the 13 candidate miRNAs have been described as biomarkers of HF: miR-21-3p, miR-21-5p, miR-23a-3p, miR-29b-3p, miR-122, miR-133a, miR-145-5p, miR-222-3p and miR-320a [10,15,16,17,18,19,20], confirming their potential as targets to predict HF (Figure 1A). To prioritize candidate miRNAs with high relation specificity with LVR, we evaluated in vivo the association of the 13 candidate miRNAs with LVR after MI (Figure 1B). In LV of post-MI rats, we selected 6 candidate miRNAs up-regulated by LVR at 7 days (miR-23a-3p, miR-222-3p and miR-320a) and at 2 months after MI (miR-21-3p, miR-21-5p, miR-222-3p and miR-377-5p) [10] and we excluded the 7 candidate miRNAs which were not detected (miR-122 and miR-210) or not modulated by LVR (miR-29b-3p, miR-133a, miR-145-5p, miR-338-3p and miR-483-3p). In plasma of post-MI rats, we selected 3 candidate miRNAs down-regulated by LVR at 7 days and up-regulated by LVR at 2 months (miR-21-5p, miR-23a-3p and miR-222-3p) [10] and we excluded the 3 candidate miRNAs which were not detected (miR-21-3p and miR-377-5p) or not measurable (miR-320a). In plasma of post-MI patients from the REVE-2 study, we showed that the 3 selected candidate miRNAs: miR-21-5p, miR-23a-3p and miR-222-3p are down-regulated by LVR at baseline and are up-regulated by LVR after 3 months, especially in men, validating their potential as circulating biomarkers of adverse LVR after MI to predict HF as previously published [10].

### 2.2. Analysis of the REVE-2 Network Identified MiR-21-5p, MiR-222-3p, MiR-335-5p, MiR-26b-5p, MiR-375 and MiR-17-5p to Detect LVR after MI

The REVE-2 molecular data generated by the measurement of 24 variables in the plasma of the patients from the REVE-2 study (including miR-21-5p, miR-23a-3p and miR-222-3p) and the EdgeLeap’s knowledge platform EdgeBox were used to build the REVE-2 molecular interaction network described in detail elsewhere [14]. The REVE-2 network contains 1310 molecules, including 24 miRNAs which were prioritized to identify candidate miRNAs to detect LVR after MI (Figure 2).

Fourteen out of the 24 candidate miRNAs are described to be associated with HF: miR-21-5p, miR-222-3p, miR-423-5p, miR-26b-5p, miR-23a-3p, miR-744-5p, miR-133a-3p, miR-17-5p, miR-29c-3p, miR-145-5p, miR-29b-3p, let-7g-5p, miR-143-3p and miR-451a [10,15,17,18,21,22,23,24], confirming they are interesting targets to predict HF (Figure 2A,B). To prioritize candidate miRNAs with high relation specificity with LVR, we evaluated in silico the association of the 24 candidate miRNAs with LVR after MI through 2 criteria: active modules and betweenness centrality (Figure 2A). To avoid the selection of miRNAs associated with mechanisms not specific to LVR such as inflammation [14], we excluded the 15 candidate miRNAs active only at baseline. However, it could be interesting to analyze the miRNAs that are only active at baseline as a potential biomarker of early LVR. To avoid the selection of miRNAs that are less significant in LVR, we excluded 13 miRNAs that were not in the top 50 molecules with the highest centrality. When combining these 2 criteria, among the 24 candidate miRNAs, only 6 remained: miR-21-5p, miR-222-3p, miR-335-5p, miR-26b-5p, miR-375 and miR-17-5p (Figure 2B). Interestingly, the 6 remaining candidate miRNAs are predicted to be active at least at 3 months after MI, when only specific mechanisms of LVR seem to be effective [14]. To date, only miR-21-5p and miR-222-3p, also identified by the first approach, were validated in vivo to detect LVR after MI [10], testifying that the signature defined by the 6 last candidate miRNAs may be used as a circulating biomarker of adverse LVR after MI to predict HF.

### 2.3. Gene Ontology Analysis of the 7 MiRNAs Targets Predicted with a High Relation Specificity with Processes Involved in LVR after MI

An analysis of the 7 miRNAs targets was performed using the ClueGO [25] and CluePedia [26] applications of Cytoscape (version 3.4.0). The applications used the miRecords database to identify the experimentally validated targets of each miRNA which were submitted to Gene Ontology enrichment (Figure 3).

Thirty-one targets were predicted to interact with 5 out of 7 miRNAs (miR-21-5p, miR-222-3p, miR-23a-3p, miR-375 and miR-17-5p) (Figure 3A). The 2 remaining miRNAs (miR-335-5p and miR-26b-5p) had no validated target in miRecords. We can observe that the miRNA’s targets are involved in pathways involved in LVR development such as fibroblast proliferation, regulation of reactive oxygen species metabolism and intrinsic apoptotic signaling pathways, but also embryonic heart development as embryonic heart tube development and aorta development (Figure 3B). These results indicate the involvement of the 7 miRNAs in cardiac development and LVR processes.

## 3. Discussion and Perspectives

Systems biology approaches have been shown to improve the biomarker discovery in comparison with traditional omics approaches by allowing the selection of biomarkers with a biological relevance in the pathology [5,6]. In this study, we described two systems biology analyses integrating both omics data to identify candidate miRNAs associated with LVR after MI and to predict HF. As interaction networks rely on the use of public databases, the prediction of miRNA interactions is not always experimentally validated and may give an approximate vision of the complex mechanisms underlying LVR [27,28]. To avoid this issue, predicted miRNA interactions were either experimentally validated or predicted by at least 3 different databases in this study. Moreover, while most systems biology approaches often integrate a single timepoint, we used different timepoints to integrate the molecular events underlying LVR progression. Finally, both approaches include a selection of candidate miRNAs associated with LVR after MI either by experimental validation in vivo or by in silico priorization before validation in post-MI patients.

The first approach, derived from a post-MI rat model, predicted 13 candidate miRNAs and 3 of these candidate miRNAs were selected to be associated with LVR in vivo both in LV and in plasma of post-MI rats: miR-21-5p, miR-23a-3p and miR-222-3p. In this first approach, candidate miRNAs were selected based on the hypothesized behavior of circulating miRNAs as an endocrine signal, but our results show that candidate miRNAs do not necessarily have the same profile in LV and in plasma, showing that miRNAs secreted by LV are quantitatively and qualitatively different from their intracellular profile in LV. Moreover, although circulating miRNA profile in HF rat models may not reflect a human profile [29], we validated the association of miR-21-5p, miR-23a-3p and miR-222-3p with LVR after MI both in plasma of the HF rat model and in HF patients, indicating that these 3 selected miRNAs define a conserved miRNA signature of LVR. The second approach predicted 24 candidate miRNAs among 1310 molecules and 6 of these miRNAs were selected to be associated with LVR in silico: miR-17-5p, miR-21-5p, miR-26b-5p, miR-222-3p, miR-335-5p and miR-375. Although miR-21-5p and miR-222-3p were validated by the first approach, miR-335-5p, miR-26b-5p, miR-375 and miR-17-5p remain to be validated. Altogether, these two integrative systems biology analyses identified a signature of 7 microRNAs associated with LVR after MI and more especially with mechanisms underlying LVR, such as apoptosis, oxidative stress and fibroblasts proliferation.

MiR-21-5p is one of the most studied miRNAs in cardiovascular diseases. Indeed, it was shown that the failing heart was able to release miR-21-5p into the circulation [18]. Mir-23a-3p is less studied. However, a correlation was described between miR-23a and pulmonary function of patients with idiopathic pulmonary hypertension [30]. MiR-23a was also shown to regulate cardiomyocyte apoptosis by targeting SOD2 mRNA [31]. MiR-222-3p is known to regulate SOD2 expression in HF patients [10] and to be involved in the inhibition of myocardial fibrosis by targeting TGFβ [32]. Recently, we showed that miR-21-5p, miR-23a-3p and miR-222-3p were decreased in the plasma of patients with high LVR at baseline and increased at one year [10]. MiR-335-5p has not been yet linked with cardiovascular diseases and it has been described as a potential biomarker of osteosarcoma in children [33] and of osteoporosis [34]. MiR-375 has been shown to be mainly expressed in the developing heart and to a lower extent in the adult heart [35] and it is also increased in the plasma of pregnant women with fetal congenital heart defects [36]. MiR-26b-5p and miR-17-5p are well known to be involved in the cardiovascular system. MiR-26b-5p has been shown to be modulated in plasma of patients presenting major cardiovascular events [23]. MiR-17-5p was shown to be increased in the plasma of patients with hypertrophic cardiomyopathy and diffuse myocardial fibrosis [24]. These results testify that miR-21-5p, miR-23a-3p, miR-222-3p, miR-320a, miR-335-5p, miR-26b-5p, miR-375 and miR-17-5p may be interesting targets to predict HF.

In conclusion, we highlighted the use of integrative systems biology analyses to define a miRNA signature of LVR to predict HF. However, even though the building of molecular networks relies on experimental data, we showed that miRNA signatures still need to be experimentally validated to be relevant in vivo.

Our study has some limitations: further experiments should be done in order to understand why there is no correlation for some miRNAs between the intracellular (LV) and extracellular (plasma) miRNA profiles. It would be important to determine the cellular origin (fibroblasts, cardiomyocytes, endothelial cells) of miRNAs and the characterization of their transporters from the intracellular to extracellular medium.

## 4. Methods

### 4.1. Experimental Model of HF in Rats

All animal experiments were performed according to the Guide for the Care and Use of Laboratory Animals published by the US National Institutes of Health (NIH publication NO1-OD-4-2-139, revised in 2011). Experimental protocols were performed under the supervision of a person authorized to perform experiments on live animals (F. Pinet: 59-350126 and E. Dubois-Deruy: 59-350253). Approval was granted by the institutional ethics review board (CEEA Nord Pas-de-Calais N°242011, January 2012). MI was induced as previously described [10,11] by permanent left anterior descending coronary artery ligation in 10 weeks old Wistar male rats. Hemodynamic and echocardiographic measurements were performed to detect LVR, at 7 days (n = 8, respectively in sham- and MI-rats) and 2 months (n = 16, respectively in sham- and MI-rats) after surgery, followed by non-infarcted area of LV sampling and plasma sampling.

### 4.2. The REVE-2 Study

The REVE-2 study is a prospective multicenter study to analyze the association between circulating biomarkers and LVR and has been previously detailed [3]. Briefly, REVE-2 study included 226 patients with a first anterior wall Q-wave MI between February 2006 and September 2008. The research protocol was approved by the Ethics Committee of the Centre Hospitalier et Universitaire de Lille, and each patient provided written informed consent. Serial echocardiographic studies were performed at hospital discharge (baseline), 3 months and one year after MI and were used to assess LVR. LVR was defined as a >20% increase in end diastolic volume from baseline to 1 year. Serial blood samples were taken at 4 timepoint: baseline, 1 month, 3 months and 12 months. Twenty-four molecular variables were measured in the REVE-2 plasma at 1 to 4 timepoints.

### 4.3. Quantification of Candidate MiRNAs

RNAs were extracted from non-infarcted area of LV of rats and from plasma of rats and REVE-2 patients as described elsewhere [10]. Candidate miRNAs were quantified and normalized with miR-423-3p in LV and with *Caenorhabditis*
*elegans* Cel-39 in plasma, as described elsewhere [10].

### 4.4. Functional Analysis of MiRNAs Targets

MiRNA target prediction was performed using the ClueGO (version 2.5.2 http://apps.cytoscape.org/apps/cluego) [25] and CluePedia (version 1.5.2 http://apps.cytoscape.org/apps/cluepedia) applications [26] of Cytoscape software (version 3.4.0 http://www.cytoscape.org/). Only validated miRNAs targets from the miRecords database were selected. A Gene Ontology (GO) enrichment analysis for Biological Processes was performed for all the miRNAs targets using the ClueGO application of Cytoscape software. *p*-value was set at 0.05, and corrected for multiple testing times using the Benjamini-Hochberg adjustment.

## Figures and Tables

**Figure 1 ncrna-05-00022-f001:**
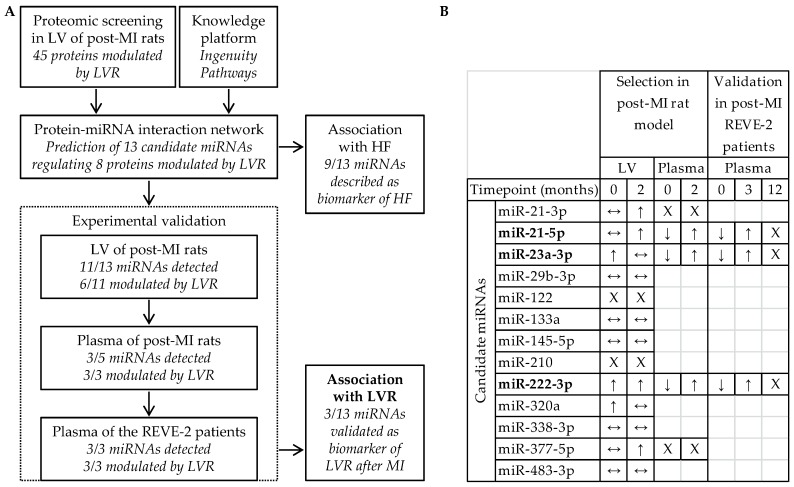
Identification of miR-21-5p, miR-23a-3p and miR-222-3p to detect left ventricular remodeling (LVR) after myocardial infarction (MI) and to predict heart failure (HF). (**A**) Design and (**B**) experimental selection and validation of the 13 candidate miRNAs predicted from the proteomic data obtained in LV of post-MI rats by the Ingenuity Pathway knowledge platform. Quantification of candidate miRNAs in LV and in plasma of post-MI rats at 7 days (Timepoint 0) and at 2 months and in plasma of REVE-2 patients at baseline (Timepoint 0), 3 month and 12 months after MI was published elsewhere [10]. ↑ and ↓ respectively indicate a significant increase and decrease of miRNAs (*p* < 0.05) detected between sham- and post-MI rats/between patients with no and high LVR, ↔ indicates no modulation, X indicates a lack of detection. MiRNAs remaining after the validation process are in bold.

**Figure 2 ncrna-05-00022-f002:**
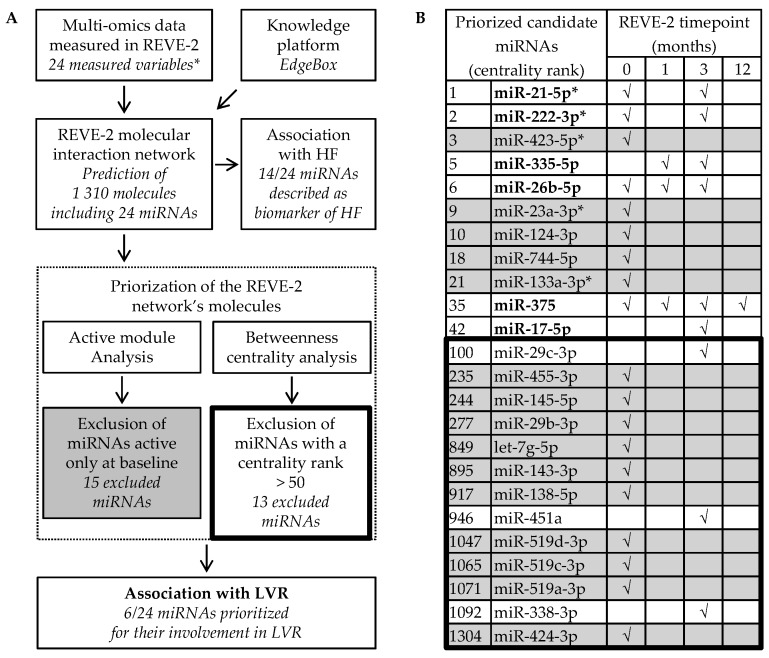
Identification of miR-21-5p, miR-222-3p, miR-335-5p, miR-26ba-3p, miR-375 and miR-17-5p to detect left ventricular remodeling (LVR) after myocardial infarction (MI) and to predict heart failure (HF). (**A**) Selection process and (**B**) priorization analysis of the 24 miRNAs predicted from the multi-omic data obtained in the 226 patients from the REVE-2 study by the EdgeBox knowledge platform. √ indicates that miRNA is predicted to be active at the corresponding timepoint: baseline (0), 1 month, 3 month and 12 months after MI. MiRNAs only active at baseline (grey) and with a betweenness centrality rank lower than 50 (inside the thick line) were excluded from further investigation because they are not expected to be highly involved in LVR after MI. * indicates REVE-2 variables. MiRNAs remaining after the selection process are in bold.

**Figure 3 ncrna-05-00022-f003:**
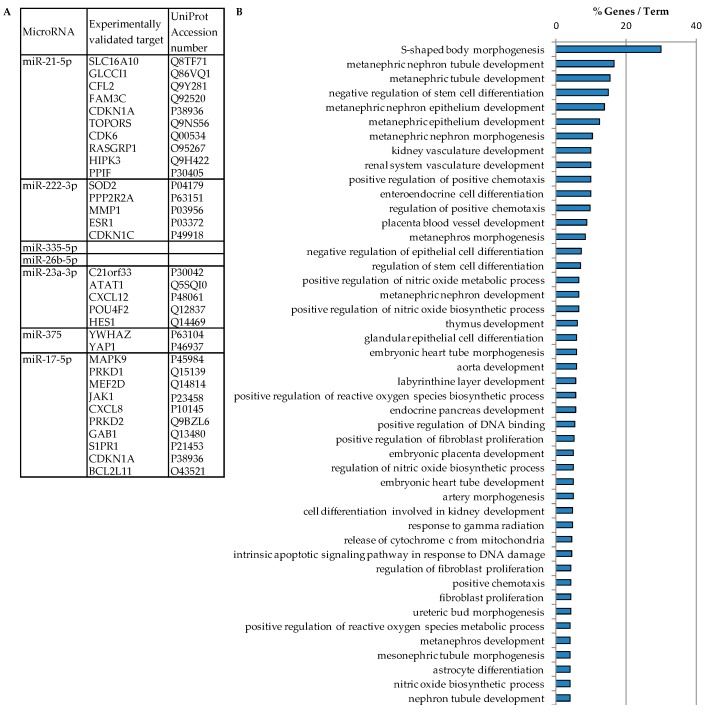
Functional annotation enrichment analysis of the 7 miRNA’s targets. (**A**) Targets of miRNAs were predicted by miRecords database. No targets have been described for miR-335-5p and miR-26b-5p. (**B**) Biological processes of miRNA’s target genes were predicted by Cytoscape plugin ClueGO and Cluepedia applications (*p* < 0.05).

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
