# Peer review of "Integrative System Biology Analyses Identify Seven MicroRNAs to Predict Heart Failure"

_ncrna, 2019, doi:10.3390/ncrna5010022_

Round 1

Reviewer 1 Report

The manuscript describes the use of two systems biology approaches to define a miRNA signature of left ventricular remodeling after myocardial infarction. By using these approaches, the authors identified a signature of 7 microRNAs associated with post-MI LVH, and three of the miRNAs were validated in vivo.

Comments

The authors list the steps of experimental validation in figure 1 and in results paragraph 2.1, but no data of this validation is shown. Similarly, all the methods of miRNA quantification from tissue and plasma samples are missing. It is therefore not possible to determine whether this validation was performed correctly.

The number of rats/samples should be reported. Were the analyzed samples from 7 day or 2 month time point? Was RNA extracted from whole left ventricle, infarcted area or non-infarcted ventricle? Which method was used for RNA extraction and qPCR in tissue and plasma samples? Was it confirmed that that there were no differences in miR-423 levels between MI and sham rats, as it was used for normalization of results? How many patient plasma samples were used, and which methods were used for RNA extraction and qPCR of these samples. What means no LVR and high LVR, how LVR was defined in REVE-2 patients? Which time point patient samples were used for qPCR? Both methods and data of this validation should be added to allow evaluation of the results.

The second approach identifies 6 miRNAs to associate with LVR, and together with approach 1 the final miRNA signature is 7 miRNA candidates. Validation of all these miRNAs in patient plasma samples would strengthen this manuscript significantly.

Discussion is very brief. For example, the role of validated miRNAs miR-21-5p, miR-23a-3p, and miR-222-3p is not discussed at all.

Author Response

The manuscript describes the use of two systems biology approaches to define a miRNA signature of left ventricular remodeling after myocardial infarction. By using these approaches, the authors identified a signature of 7 microRNAs associated with post-MI LVH, and three of the miRNAs were validated in vivo.

Comments

The authors list the steps of experimental validation in figure 1 and in results paragraph 2.1, but no data of this validation is shown. Similarly, all the methods of miRNA quantification from tissue and plasma samples are missing. It is therefore not possible to determine whether this validation was performed correctly.

We apologize and have modified the revised manuscript to clarify. Aspart of experimetns have been already published, we have reference it instead to duplicate work already done.

The figure 1 was modified to detail the validation step

The number of rats/samples should be reported. Were the analyzed samples from 7 day or 2 month time point? Was RNA extracted from whole left ventricle, infarcted area or non-infarcted ventricle? Which method was used for RNA extraction and qPCR in tissue and plasma samples? Was it confirmed that that there were no differences in miR-423 levels between MI and sham rats, as it was used for normalization of results? How many patient plasma samples were used, and which methods were used for RNA extraction and qPCR of these samples. What means no LVR and high LVR, how LVR was defined in REVE-2 patients? Which time point patient samples were used for qPCR? Both methods and data of this validation should be added to allow evaluation of the results.

We have detailed in the methods section information about the number of rats and the number of patients. It is now stated that we have analyzed the non-infarcted area.

We have not detailed the protocole of RNA extraction and quantification as it was published recently but we have provided the reference of the paper and the normalizers used.

We have explained in the paragraph concerning the REVE2 study how LVR is defined.

In the new figure 1, all the time points are indicated

The second approach identifies 6 miRNAs to associate with LVR, and together with approach 1 the final miRNA signature is 7 miRNA candidates. Validation of all these miRNAs in patient plasma samples would strengthen this manuscript significantly.

We agree with the reviewer but before doing the analyzing in patients we would to end up the evaluation of all the miRNAs detected to be modulated in LV in plasma of the experimental model

Discussion is very brief. For example, the role of validated miRNAs miR-21-5p, miR-23a-3p, and miR-222-3p is not discussed at all.

In the revised manuscript, the discussion is extended and the 3 miRNAs validated are discussed. We had additional references

Reviewer 2 Report

Charrier et al suggest an interesting approach for the identification of novel miRNAs as biomarkers of HF. I have some minor comments:

The manuscript needs a Limitations paragraph in the Discussion section. The authors should discuss in detail the poor understanding of circulating miRNA biology. Indeed, their approach is based on the hypothesized behavior of circulating miRNAs as endocrine mediators or endocrine genetic signals. However, until now this has not been probed. The correlation between the intracellular extracellular miRNA profiles from the same cell/tissue is not clear, which could limits the conclusions of the study. In case the miRNAs secreted by LV are quantitatively and qualitatively different from the intracellular profile of the tissue, the translations of the results presented could be affected.  In addition, in silico approaches and bioinformatic prediction tools have a number of limitations that should be carefully discussed. This will complement their excellent work.

Additional information about the experimental model in rats and functional analysis is fundamental.

Author Response

Charrier et al suggest an interesting approach for the identification of novel miRNAs as biomarkers of HF. I have some minor comments:

The manuscript needs a Limitations paragraph in the Discussion section. The authors should discuss in detail the poor understanding of circulating miRNA biology. Indeed, their approach is based on the hypothesized behavior of circulating miRNAs as endocrine mediators or endocrine genetic signals. However, until now this has not been probed. The correlation between the intracellular extracellular miRNA profiles from the same cell/tissue is not clear, which could limits the conclusions of the study. In case the miRNAs secreted by LV are quantitatively and qualitatively different from the intracellular profile of the tissue, the translations of the results presented could be affected.  In addition, in silico approaches and bioinformatic prediction tools have a number of limitations that should be carefully discussed. This will complement their excellent work.

We added a limitation praragraph at the end of the discussion

Additional information about the experimental model in rats and functional analysis is fundamental.

We detailed information on the methods section about the experimental model, patients and process of extraction and quantification. But as it was previously  published, we summarize and referenced the paper.

Round 2

Reviewer 1 Report

The authors have made suggested corrections to the manuscript or provided adequate responses to questions and comments.

Reviewer 2 Report

The authors have addressed my comments.